# Validity of the Swimming Capacities and Anthropometric Indices in Predicting the Long-Term Success of Male Water Polo Players: A Position-Specific Prospective Analysis over a Ten-Year Period

**DOI:** 10.3390/ijerph19084463

**Published:** 2022-04-07

**Authors:** Goran Dimitric, Dean Kontic, Sime Versic, Tijana Scepanovic, Natasa Zenic

**Affiliations:** 1Faculty of Sport and Physical Education, University of Novi Sad, 21000 Novi Sad, Serbia; dimitrg@gmail.com (G.D.); tijanascepanovic021@gmail.com (T.S.); 2University of Dubrovnik, 20000 Dubrovnik, Croatia; dkontic11@hotmail.com; 3Faculty of Kinesiology, University of Split, 21000 Split, Croatia; sime.versic@kifst.eu

**Keywords:** performance, sport-specific tests, generic tests, prediction, success

## Abstract

Long-term predictors of success in water polo are rarely reported. This study aimed to evaluate the position-specific validity of the swimming and anthropometric/body build tests on the prediction of the long-term success of male water polo players. The participants were 85 top-level players observed at baseline (junior level, when they were 17–18 years of age; 35 centers and 50 perimeter players) in tests of swimming capacities (15 m water polo sprint, 25 m sprint, and 100 and 400 m freestyle swimming) and anthropometric/body build indices (body mass, body height, arm span, body mass index, and body fat percentage). Over a period of 10 years, participants’ senior-level success was prospectively observed. Multinomial logistic regression with three levels of criteria (high achievement, medium achievement, and low achievement at the senior level) was applied to identify the predictive validity of the swimming and anthropometric/body build indices. For the total sample (not dividing perimeter from center players), all observed swimming capacities, body height, and arm span were significantly associated with success, with more successful players being taller, with a longer arm span, and superior swimming capacities. Body height and 100 m freestyle swimming were significant predictors of success among centers. Achievement in 100 and 400 m freestyle swimming, body height, and a lower percentage of body fat were correlated with higher senior-level performance in perimeter players. The results showed better validity of the anaerobic-lactate and aerobic swimming tests than sprint swimming in predicting the long-term achievement of male water polo players. Differences in the influence of swimming capacities and body-build indices on senior-level success between observed playing positions are related to differences in game duties, playing time, and consequent metabolic demands.

## 1. Introduction

Water polo is a team sport played between two teams, each with six field players and a goalkeeper. The origins of water polo can be found in the mid-nineteenth century, and it has been part of the Olympic program since the third Modern Olympic Games in St. Louis in 1904 [1]. Although it is played predominantly on the European continent, its growing popularity in recent years has been recorded in other parts of the world, especially in the USA and Australia [2]. Like other team sports, water polo is a polystructural activity with an intermittent character and consists of offensive and defensive phases and negative and positive transition phases [3,4]. Since the medium in which players move is water, water polo is a combination of swimming skills and abilities, combined with specific technical and tactical tasks (i.e., passing, shooting, and movements with and/or without the ball) in a constantly physical game expressed through duels and tackles [2,5,6]. Due to these specific demands to which players are exposed, water polo is considered an extremely physically demanding sport with many strength and conditioning components important for optimal performance [5].

Aerobic and anaerobic swimming endurance, swimming speed, and static, repetitive, and maximal strength and power are considered crucial factors for achieving situational efficiency and, eventually, competitive success [7,8,9,10]. Specifically, a study conducted with junior water polo players identified sport-specific fitness tests, such as a 20 m swimming sprint, an in-water jump, a drive–shoot test, a multilevel swimming endurance test, and a dynamometric semi-tethered force test, as important determinants of success [11]. Another study of a similar sample noted drive–shoot speed as the only significant predictor of qualitative level [12]. Finally, a recent study of cadet-level players revealed that national-level players achieved significantly better results on swimming tests in comparison to their team-level peers [13].

Apart from conditioning capacities, anthropometric indices are also considered important determinants of success in water polo. Namely, due to the contact nature of the game, body mass is considered an important factor of achievement for certain playing positions (i.e., center players) [5,14]. Meanwhile, greater body height and longer length of limbs will allow players to reach for the ball more easily, to shoot, and to perform blocks more efficiently. However, there is no clear consensus on the importance of anthropometric indices on performance level in water polo. In accordance with the previous hypothesis of the importance of body length dimensions, body height was highlighted as a significant qualitative discriminating factor in several studies on youth water polo players [11,13]. However, no anthropometric differences were noted in the study comparing water-polo-specific fitness status between junior-level players in the national squad and the club-level players [12].

From the previous literature overview, it is clear that studies have frequently evaluated the associations that may exist between fitness capacities, anthropometric indices, and water polo performance. However, all the mentioned studies were cross-sectional and evaluated associations between certain fitness and anthropometric attributes and water polo performance at a certain moment either by distinguishing the performance levels alongside the studied capacities [12,15] or by correlating independent variables of anthropometric/fitness status with specific indices of water polo performance [5]. On the other hand, there is an evident lack of research into the predictive validity of certain indices on future water polo competitive achievement/success. In other words, there is a clear need to explore how fitness and anthropometric attributes influence future competitive efficiency and the level of career success the players can achieve in water polo. There is no doubt that the most obvious reason for the lack of this kind of research is its duration. In other words, predictive attributes should be evaluated earlier in one’s career, and it takes time to properly evaluate players’ achievements [16].

Indeed, only few studies on this topic have been conducted in team sports [17,18]. In a study of college-level U.S. football players, the authors showed that the existing battery of tests (40-yard sprint, broad jump, high jump, bench press, 20 yards, lane agility) used in the selection of young players at their entrance in National Football League was not a good predictor of their future performance, with the exception of sprint tests for running backs [18]. Meanwhile, a study conducted with basketball players showed that lower-body power and upper-body strength had value for predicting the future performance of players [17]. To the best of our knowledge, only one study evaluated the problem of long-term prediction in water polo. Specifically, almost 20 years ago, Israeli authors studied young water polo players aged 14–15 years, with a follow-up after a 2-year period, before selection to the junior national team [19]. Their results showed that the selected players were already superior in baseline measurements in most swimming tests and also dribbling and game intelligence parameters [19]. Despite its importance, considering the length of time and the changes to water polo rules and game demands, the applicability of those findings is limited. 

Evidently, the problem of long-term development and the prediction of success in team sports is a topic worthy of attention. Although the swimming capacities and anthropometric/body build indices appear to be important determinants of water polo performance, it is not known if those indices predict the future performance of players (e.g., predictive validity). Specifically, we found only one (relatively old) study, where the authors investigated the predictive validity of certain attributes on future water polo success [19]. Finally, there is an evident lack of investigations that evaluate the predictive validity of anthropometric and fitness indices on later sport achievement in countries that are known to be highly successful in water polo. Therefore, the aim of this study was to determine the validity of swimming capacities and anthropometric/body-build indices (predictors) in predicting the future performance (success) of male water polo players. Namely, predictors were evaluated when participants were at junior level (17–18 years of age), while their sport success was evidenced throughout the following 10 years (throughout the time they were at senior level, until the age of 27–28 years). Initially, we hypothesized that swimming capacities and the anthropometric/body build observed at the junior level would have predictive validity on senior-level sport achievement in male water polo. The findings of the study will be useful for water polo coaches to determine which indices should be valued more highly when evaluating junior-level players. 

## 2. Materials and Methods

### 2.1. Participants

In this study, we observed 85 elite youth water polo players from Croatia and Montenegro (aged 17 or 18 years). All participants were junior-level members of teams that competed in the Adriatic League, one of the strongest team-level competitions in the world. For the purpose of this study, participants were observed according to their playing positions and were divided in two groups: (i) center players (center forwards and points, *n* = 35) and (ii) perimeter players (wings and drivers, *n* = 50). At the time of the initial screening, all participants had participated in water polo training for at least 7 to 9 years, and their usual weekly exposure to training was 18–22 h, which included 8–10 training sessions and one game. Training consisted of work on both technical–tactical aspects and strength and conditioning (swimming and gym work). All participants were invited to participate in the study by their national water polo federations. Before the initial testing procedure, the examiners explained the aims, purposes, benefits, and potential risks of the investigations to the participants. All participants voluntarily agreed to participate in the measurements and signed an informed consent form themselves, or their parents signed if the participants were under 18 years old. The study was approved by the Ethics Committee of the University of Split, Faculty of Kinesiology, and was conducted in accordance with the Code of Ethics of the World Medical Association.

### 2.2. Variables and Study Design

Apart from the playing position, which was recorded during the initial testing, in this study, we observed predictors and a criterion. The predictors were anthropometric/body composition indices and the tests of swimming capacity; these were measured when players were at the junior level (initial screening). As the criterion, we observed players’ competitive achievement over a period of eight years after the initial screening (Figure 1).

Anthropometric/body-build indices included body height, body mass, body mass index (BMI), arm span, and body fat percentage (BF%). Body height was expressed in centimeters and measured with an anthropometer (Seca, Birmingham, UK), while Tanita scales (TBF-300, Tanita, Tokyo, Japan) were used for body mass measurement (kilograms), and these data were used for BMI calculation (kg/m^2^). For the BF% calculation, we used the Durnin and Wormersley method [20]. First, biceps, triceps, and subscapular and suprailiac skinfolds (SF) were measured with calipers (Holtain, London, UK). These values were used for body density (BD): BD = 1.162 − 0.063 × log Σ4SF. Finally, the BF% was calculated from the BD with following formula: BF% = (4.95/BD − 4.5) × 100.

The swimming tests were 25, 100, and 400 m freestyle swimming (25 mFS, 100 mFS, and 400 mFS, respectively) and 15 m of specific water-polo-sprint swimming (15 mWS). All tests were performed in a 25 m long swimming pool, and a Longines swimming timing apparatus was used for time measurement (0.01 s). For the 25 mFS, 100 mFS, and 400 mFS, players started from the water and were allowed to push themselves from the wall (i.e., they did not jump from the starting block). For the 15 mWS, the participants started from the pool, 15 m from the finishing block, and they could not push themselves from the wall. During the 15 mWS, players had to maintain a specific water polo position with the head remaining out of the water for the entire distance [21]. The 100 mFS and 400 mFS were performed once, while the 25 mFS and 15 mWS were performed three times with a rest of 3–4 min between each test. After the reliability analysis (intraclass correlation = 0.78 and 0.81 for 25 mFS and 15 mWS, respectively), the best achievement was used for further analysis. 

The criterion (senior-level performance) was observed throughout the following 10 years after the initial screening (follow-up period). For the purpose of this study, we categorized players into three groups according to their highest achievement at the senior level. The first group consisted of those players who were team members of senior-level water polo teams that competed in international-level competition (G1; high achievers; *n* = 29). In the second group, we clustered those players who competed at the senior level in teams that competed in the highest national playing division (G2; medium achievers; *n* = 26). The third group consisted of players who could not be categorized in G1 and G2 but were screened initially as G1 and G2 (low achievers, *n* = 30). In other words, players grouped into G3 were players who competed in lower-level competitions at the senior level or quit competitive water polo after the junior level.

### 2.3. Statistics

Kolmogorov–Smirnov test was applied to identify the normality of the distributions. Consequently, means and standard deviations were reported for all predictors. Levene’s test was calculated to define the equality of the variances. 

Differences among qualitative groups (high achievers, medium achievers, and low achievers) were evidenced throughout one-way analysis of variance (ANOVA) with Scheffe’s post hoc comparison. The effect size differences (ES) were evidenced on the basis of ANOVA-derived partial eta-squared values (η^2^; small ES > 0.02; medium ES > 0.13; large ES > 0.26).

In order to identify the influence of predictors (anthropometric/body-build variables and swimming capacities obtained at junior level) on criterion (trinomial criterion—success at senior level; high achievement/medium achievement/low achievement), we calculated multinomial regression, with odds ratio (OR) and 95% confidence interval (95%CI) reported. In multinomial regression, the high-achievement group was used as reference value. Statistica ver. 13.5 (Tibco Inc., Palo Alto, CA, USA) was used for all calculations, with the significance level of *p* < 0.05.

## 3. Results

Descriptive statistics for the total sample of participants are presented in Appendix A.

When analyzed for total sample of participants, ANOVA showed significant differences for all swimming capacities and body height among three qualitative groups. In brief, in their junior age, the best achievers were taller and had better swimming capacities than their less successful peers. The ES differences among groups were evidenced as being medium for body height, 25 mFS, 15 mWS, 100 mFS, and 400 mFS and small for remaining variables. In most cases, significant post hoc differences were evidenced between high achievers and low achievers, but even medium achievers performed better than low achievers in 400 mFS (Table 1).

Table 2 presents ANOVA differences among achievement groups for center players. High-achievers were taller (significant post hoc differences between all groups) and performed better in 100 mFS (significant post hoc differences between high achievers and low achievers) than their less successful peers, with large ES differences among groups for body height and 100 mFS and medium ES differences for arm span and 400 mFS.

Most successful perimeter players were taller (significant post hoc difference when compared to both remaining groups) and performed better in 100 mFS and 400 mFS (significant post hoc difference when compared to low-achievers). The ES differences among groups were evidenced as being of large magnitude for 100 mFS and 400 mFS and of medium magnitude for body height and 25 mFS (Table 3).

Multinomial regression results calculated for total sample are presented in Figure 2. Body length dimension at junior age was a factor that significantly contributed to success at senior level. In brief, lower likelihood for being categorized in G2 and G3 was found for taller players (OR = 0.81, 95%CI: 0.72–0.91, and OR = 0.78, 95%CI: 0.67–0.91 for G1 and G2, respectively). These results are logically accompanied by significant contribution of arm-span to success of the players at their senior age (G1: OR = 0.91, 95%CI: 0.85–0.98) (Figure 2A). Performance in 25 mFS (G1: OR = 3.71, 95%CI: 1.65–5.36), 15 mWS (G1: OR = 1.5, 95%CI: 1.01–2.05), 100 mFS (G1: OR = 1.64, 95%CI: 0.29–2.10), and 400 mFS (G1: OR = 1.11, 95%CI: 1.05–1.16) evidenced at junior age contributed significantly to success of water polo players at their senior age, with poorer results in low achievers when compared to reference group (high achievers) (Figure 2B).

Figure 3 presents results of the multinomial regression calculated for centers. Lower likelihood for achieving success at senior age was evidenced for centers who had shorter body lengths in their junior age (G1: OR = 0.74, 95%CI: 0.59–0.91; OR = 0.87, 95%CI: 0.76–0.98, for body height and arm span, respectively) (Figure 3A). Junior-level result at 100 mFS significantly contributed to success at senior level, with higher likelihood for being a low achiever among those players who performed poorly at this swimming capacity (OR = 1.73, 95%CI: 1.14–2.63) (Figure 3B).

Multinomial regression results for perimeter players are presented in Figure 4. Body height is a significant predictor of senior-level success (G1: OR = 0.79, 95%CI: 0.65–0.95; G2: OR = 0.74, 95%CI: 0.59–0.93). Higher likelihood for being categorized as a low achiever (OR = 1.28, 95%CI: 1.02–1.62) and medium achiever (OR = 1.39, 95%CI: 1.03–1.89) as a senior-level player was evidenced for those participants who had higher BF% at their junior age (Figure 4A). Those players who did not achieve high success at their senior level did not perform well at 100 mFS (G1: 1.14, 95%CI: 1.06–1.29) and 400 mFS (G1: 1.94, 95%CI: 1.24–2.93; G2: OR = 1.57, 95%CI: 1.06–2.23) in their junior age (Figure 4B).

## 4. Discussion

With regard to study aims, there are three most important findings. First, for the total sample (i.e., not dividing players according to their playing position), body lengths (body height and arm span) and all screened swimming capacities were found to be important determinants of future success. Second, centers who were evidenced as being more successful at the senior level were taller and possessed better swimming capacities at the junior level than their peers who did not achieve competitive success at the senior level. Third, superior swimming capacity at 100 mFS and 400 mFS, body height, and a lower percentage of body fat were correlated with superior senior-level performance in perimeter players. Therefore, since swimming capacities and anthropometric/body-build indices observed at the junior level were found to be valid predictors of success at the senior level, our initial study hypothesis can be accepted. In the following discussion, the main findings will be discussed separately for swimming capacities and anthropometric/body-build indices.

### 4.1. Swimming Capacities and Success in Water Polo

When we observed the total sample of water polo players (not divided according to playing position), all swimming capacities at the junior level were found to be valid predictors of players’ achievement at the senior level. Given that swimming is a major component of water polo, these results were not surprising and were generally in accordance with previous findings. For example, a study of junior-level water polo players evidenced the dominance of the national team players over their club-level peers in aerobic endurance (a multistage swimming test) and sprint-swimming capacity (20 m sprint) [11]. Likewise, a study with a sample of younger players (average 14.34 ± 0.25 years of age) found significant differences in all applied swimming tests (800, 400, 100, 50, and 25 m freestyle and 25 m water polo sprint) between qualitatively different groups of players [13]. Similarly, a study of a sample of players at the same age described multiple swimming freestyle tests as adequate selection criteria in youth water polo categories [19].

Indeed, water polo is a highly intensive sport with a significant contribution of both aerobic and anaerobic metabolism [5]. An analysis showed that the average distance covered in a match is 1.613 ± 150 m, with 44% covered at high intensity (faster than 1.4 m/s) and average blood lactate levels of 7.7 ± 1.0 mmol/L [7]. However, due to positional specifics and in-game roles, energy demands are different between center and perimeter players, which can be seen in blood lactate levels that vary from 5.3 to 11.2 mmol/L [7]. Therefore, for a deeper analysis of the influence of swimming capacities on success in water polo, a position-specific approach is more applicable.

When the contribution of swimming capacities to competitive achievement are analyzed by playing positions, it seems that sprint swimming tested at the junior level does not contribute to players’ success at the senior level. On the other hand, the longer swimming tests have significant predictive validity. In particular, the highest predictive value for centers was the 100 mFS. Meanwhile, the 400 mFS was the strongest predictor of senior-level success for perimeter players. These differences between the contributions of swimming capacities to achievement for playing positions can be explained by three positional specific factors: (i) playing time, (ii) distance covered, and (iii) game load.

Regarding the specific field position, perimeter players need to cover longer distances in the game than center players. This is particularly evident in transition (i.e., when they perform transitions between offense and defense) and in power play situations (i.e., where they need to be more agile and mobile in defensive duties) [12,15]. Perimeter players are less frequently substituted during the game and therefore have more playing time [22]. This is partially influenced by their game load, as they are rarely involved in intensive wrestling and fighting for position (i.e., the contact game), which is specific to centers. Such differences in game duties, together with the larger body mass of centers, causes higher metabolic loads, which is clearly evidenced in higher concentrations of blood lactates in center players compared to perimeter players [7].

The previous discussion leads us to the conclusion that the generic 100 m swimming test (100 mFS) can be used as good indicator of anaerobic swimming capacity in centers despite the test’s simplicity. On the other hand, the performance of perimeter players is strongly related to their aerobic capacities, which is the most important factor in the 400 mFS. Although we did not find any studies with a prospective analysis over a relatively long period of time for water polo, our findings can be indirectly supported by previous cross-sectional studies. In brief, when observed at the junior level, players who were members of the national team had significantly better results on the aerobic swimming test than their peers at a lower performance level [11].

It is interesting to note that position-specific analyses did not reveal sprint-swimming tests as significant predictors of the future success of the observed players. Indeed, this is in contrast to some previous research, where the authors highlighted sprint tests (15 and 25 m) as good predictors of qualitative level in youth [11,13]. However, we must note that the previous studies were cross-sectional and explored relations between swimming capacities and water polo quality at a particular moment in time. On the other hand, we evaluated predictors of future success. We can therefore assume that the sprint-swimming tests are good selection criterion for youth players, but the player’s eventual success will be more related to specific aerobic or anaerobic capacities depending on position as previously discussed. However, it is also possible that the changes in the official water polo rules that have occurred over time also contributed to our findings. In brief, changes in the official water polo rules might impact the overall physiological and metabolic demands of the game, which could (potentially) result in increased aerobic and (lactate) anaerobic demands while reducing the sprint swimming demands of the sport in general.

### 4.2. Anthropometric Indices, Body Build, and Success in Water Polo

For the total sample, body height and arm span were indicated as valid predictors of future water polo performance. As these two variables present the same anthropometric dimension (i.e., body lengths), interpretation of their influence on performance is straightforward. In general, the positive influence of body lengths on senior-level achievement was expected, as it is well known that body size is a major factor contributing to water polo success [11]. Generally, taller players with longer arms can reach further for the ball, have better control of passes and shots, as well as control over an opponent in fighting and wrestling for the optimal position [8,23].

Specifically, the kinematic analysis of water polo showed that approximately more than half of the game time is spent in so-called vertical swimming, where a player’s height can be a crucial factor for efficiency [24,25,26]. The importance of body size in water polo has been confirmed in numerous studies regardless of age and gender [11,23]. A study of elite Australian female senior water polo players showed that the National Squad players were significantly taller than the National League players [23]. This is also important in youth water polo, as studies with junior- and cadet-level players reported height as an important factor that distinguished players at different qualitative levels [11,13]. Given that the players in the observed sample were in the in the final phase of growth and development at the time of the initial testing (17–18 years of age), it is clear that their body size played a significant role in the process of selection between the junior and senior categories. In other words, taller junior players probably have a higher chance of success at the highest (senior) level.

Body height is a significant predictor of players’ quality even when stratified for playing positions. Although center players are generally taller than perimeter players, the overall body height is important for both positions [25,27]. Considering the aforementioned vertical component of water polo, the importance of the body height is manifested partially in the same way in both groups of players. However, there are also some position-specific roles that need to be highlighted when analyzing these results.

As already discussed, center players (center forwards and points) spend the majority of their time in direct confrontation with the opponent both during defense and offense. In these situations, body height can help them in keeping distance from the opponent and reaching for the ball more easily [27]. On the other hand, perimeter players are not so involved in the contact game during the offense, but they pass the ball around the defense zone and shoot from a distance. Meanwhile, in defense, they have to block or intercept opponents’ shots or passes in order to protect the goal [25]. Consequently, taller players will have a certain advantage in these situations.

The significant predictor of achieved qualitative level for perimeter players was BF%, and players with a lower BF% at the junior level were more successful at the senior level. In general, perimeter players record lower BF% values than centers, and this finding is explainable by the position-specific game tasks [27]. As mentioned, perimeter players cover more swimming distance and have more playing time compared to centers [22]. In these activities, body fat represents ballast mass and unnecessary weight and therefore negatively affects swimming performance. Although studies report that in water sports, body fat has a positive influence on buoyancy, this does not mean that increasing the body fat level above certain (necessary) values is desirable although the normative values for body fat in water sports (i.e., water polo and artistic swimming) are generally higher compared to other athletes [11,28,29]. A study of 110 elite junior-age players indicated that while centers had the highest levels of BF%, the players in the perimeter positions recorded the lowest values [27]. Similar findings occurred in a study of senior-level female players, with perimeter players having significantly less adipose tissue [23]. Therefore, we can conclude that, regardless of age and gender, lean body mass is an important anthropometric factor for competitive efficiency as a perimeter player.

### 4.3. Strengths and Limitations of the Study

The unequal number of participants in the observed playing-position groups is one of the study limitations. However, this was due to the fact that there are usually four perimeter and two central players in a standard lineup. Additionally, the conducted physical tests only included sprint, anaerobic, and aerobic swimming parameters. There are several more strength and conditioning capacities, such as strength, power, agility, and precision, that should be observed in future studies to obtain a more detailed picture of the predictors of future success in water polo. As a methodological remark, we must mention that there is a possibility that some other statistical approaches were also applicable (i.e., dividing players into only two performance groups and consequent calculation of the logistic regression), but we were of the opinion that the applied clustering into three groups and calculation of the multinomial regression allowed us to identify the “real-sport” indicators of success more specifically.

One of the major strengths of the study was the quality of the sample, as it consisted of elite young players who were members of teams competing in one of the strongest competitions in the world, including members of the youth national teams of the world’s best national selections. Moreover, as mentioned previously, this was the first study to examine the predictive validity of anthropometric and physical test for long-term success in water polo, and these findings have a high practical value for water polo coaches and practitioners.

## 5. Conclusions

The results of this study showed that anthropometric indices and swimming capacities can be considered as valid predictors of the sporting achievements of water polo players. In particular, body height seemed to be a relevant factor for both center and perimeter players, while a lower percentage of body fat was found to be crucial for the future success of perimeter players.

Swimming performance at the junior level also proved to be a significant predictive factor for future performance. In general, swimming capacities played a significant role in water polo efficiency and players’ overall achievement. However, the results were position-specific with regard to the influence of certain swimming capacities on players’ future achievement.

In particular, aerobic swimming endurance was more important for perimeter players, while anaerobic swimming capacities played a significant role in the future performance of center players. Meanwhile, sprint swimming was not found to be an important determinant of success when playing positions were stratified. Therefore, although the sprint-swimming tests were repeatedly found to be valuable selection criteria at a younger age, later success at the senior level was strongly related to position-specific energy capacities (i.e., anaerobic-lactate swimming capacity for centers and aerobic swimming capacity for perimeter players).

Considering that this study showed a proper predictive validity of the applied swimming tests, it provides good practical guidelines for water polo coaches to use in testing and selection procedures. Finally, given that the sample of participants included elite water polo players (including several senior-level Olympic medal winners), these findings can be considered as a valuable contribution to the highest level of water polo performance with practical applications.

## Figures and Tables

**Figure 1 ijerph-19-04463-f001:**
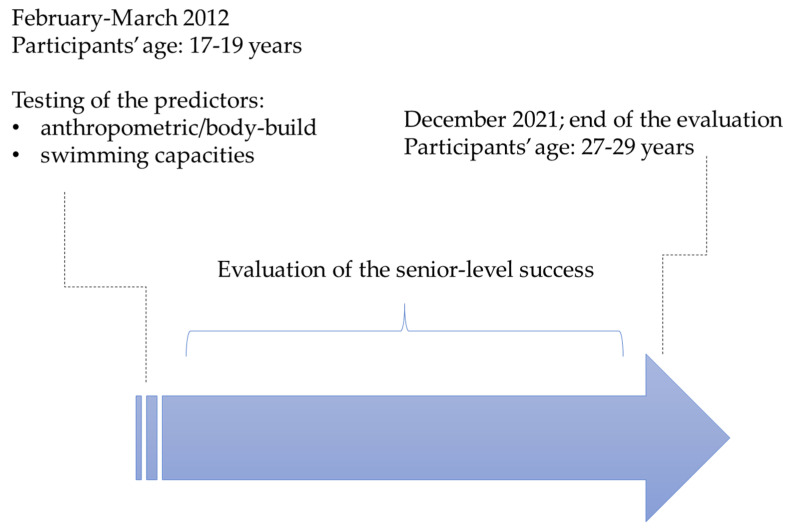
Study design.

**Figure 2 ijerph-19-04463-f002:**
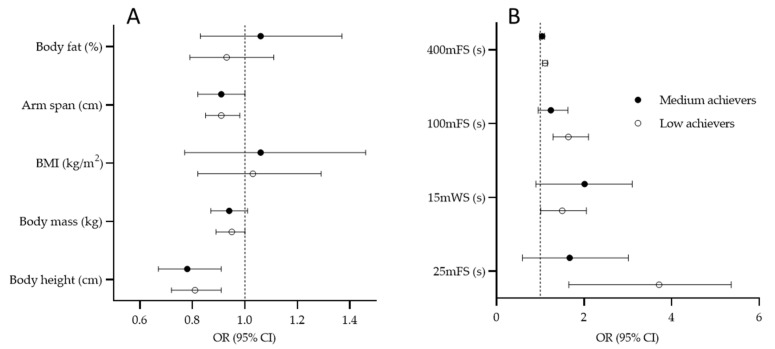
Results of the multinomial logistic regression for criterion “success at senior level” with high-achievement group as reference value for anthropometric predictors (**A**) and swimming predictors; (**B**) total sample (BMI, body mass index; 25 mFS, 25 m freestyle swimming; 15 mWS, 15 m water polo swimming; 100 mFS, 100 m freestyle swimming; 400 mFS, 400 m freestyle swimming).

**Figure 3 ijerph-19-04463-f003:**
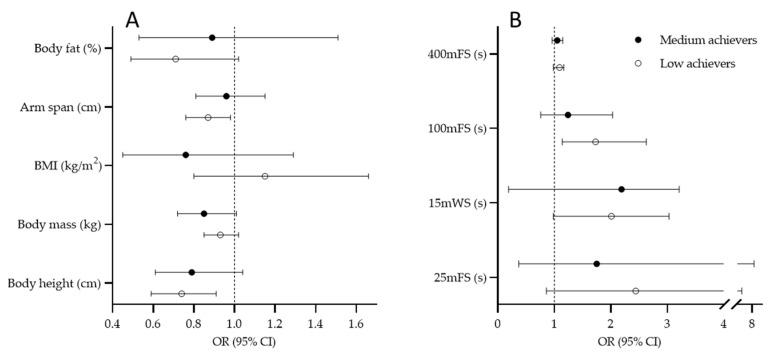
Results of the multinomial logistic regression for criterion “success at senior level” with high-achievement group as reference value for anthropometric predictors (**A**) and swimming predictors; (**B**) center players (BMI, body mass index; 25 mFS, 25 m freestyle swimming; 15 mWS, 15 m water polo swimming; 100 mFS, 100 m freestyle swimming; 400 mFS, 400 m freestyle swimming).

**Figure 4 ijerph-19-04463-f004:**
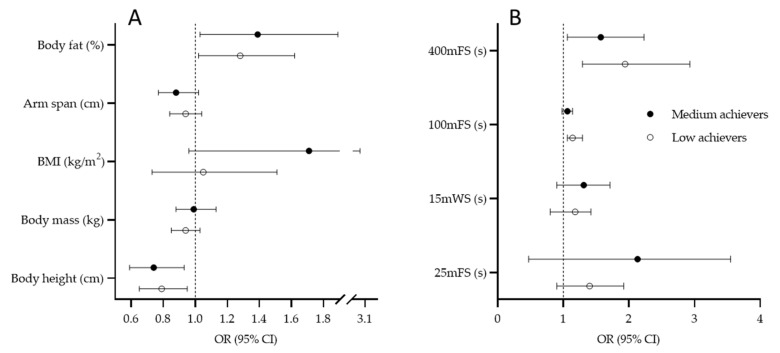
Results of the multinomial logistic regression for criterion “success at senior level” with high-achievement group as reference value for anthropometric predictors (**A**) and swimming predictors; (**B**) perimeter players (BMI, body mass index; 25 mFS, 25-meter freestyle swimming; 15 mWS, 15 m water polo swimming; 100 mFS, 100 m freestyle swimming; 400 mFS, 400 m freestyle swimming).

**Table 1 ijerph-19-04463-t001:** Descriptive statistics and differences derived by analysis of variance (ANOVA) among qualitative groups for total sample of water polo players (results are presented as means ± standard deviations).

	Low Achievers	Medium Achievers	High Achievers	ANOVA		
	*N* = 30	*N* = 26	*N* = 29	F-Test	*p*	η^2^
Body height (cm)	184.42 ± 5.02	183.67 ± 6.44	190.24 ± 5.57	11.75	0.001	0.22
Body mass (kg)	83.19 ± 9.76	82.77 ± 7.66	88.14 ± 9.64	2.71	0.07	0.06
BMI (kg/m^2^)	24.43 ± 2.35	24.57 ± 2.24	24.29 ± 1.76	0.08	0.93	0.01
Arm span (cm)	192.63 ± 6.73	192.24 ± 9.07	197.12 ± 7.04	3.88	0.02	0.09
Body fat (%)	18.17 ± 2.77	19.17 ± 3.91	18.7 ± 2.57	0.67	0.51	0.05
25 mFS (s)	13.3 ± 0.74	12.93 ± 0.64	12.7 ± 0.64	6.65	0.001	0.14
15 mWS (s)	9.21 ± 0.49	8.91 ± 0.42	8.71 ± 0.39	11.03	0.001	0.20
100 mFS (s)	63.33 ± 3.51	60.96 ± 2.57	59.56 ± 2.48	13.63	0.001	0.22
400 mFS (s)	307.68 ± 20.26	293.04 ± 13.9	286.37 ± 12.92	13.97	0.001	0.23

Legend: BMI, body mass index; 25 mFS, 25 m freestyle swimming; 15 mWS, 15 m water polo swimming; 100 mFS, 100 m freestyle swimming; 400 mFS, 400 m freestyle swimming.

**Table 2 ijerph-19-04463-t002:** Descriptive statistics and differences derived by analysis of variance (ANOVA) among qualitative groups for center players (results are presented as means ± standard deviations).

	Low Achievers	Medium Achievers	High Achievers	ANOVA		
	*N* = 11	*N* = 10	*N* = 14	F-Test	*p*	η^2^
Body height (cm)	186.97 ± 4.65	188.38 ± 5.83	193.72 ± 5.13	7.27	0.001	0.31
Body mass (kg)	90.32 ± 8.01	84.88 ± 11.49	94.84 ± 8.09	2.50	0.10	0.11
BMI (kg/m^2^)	25.84 ± 2.1	24.02 ± 3.9	25.25 ± 1.66	1.17	0.32	0.01
Arm span (cm)	195.17 ± 5.87	199.38 ± 8.48	200.96 ± 7.15	2.98	0.07	0.15
Body fat (%)	18.16 ± 2.52	19.42 ± 4.36	19.9 ± 1.35	2.10	0.14	0.07
25 mFS (s)	13.34 ± 0.84	13.17 ± 0.66	12.87 ± 0.66	1.50	0.24	0.08
15 mWS (s)	9.4 ± 0.57	9.03 ± 0.67	8.85 ± 0.99	2.20	0.11	0.08
100 mFS (s)	63.96 ± 3.17	61.86 ± 3.62	60.67 ± 2.05	5.31	0.01	0.25
400 mFS (s)	306.13 ± 19.08	298.5 ± 16.58	291.55 ± 15.58	2.56	0.09	0.18

Legend: BMI, body mass index; 25 mFS, 25 m freestyle swimming; 15 mWS, 15 m water polo swimming; 100 mFS, 100 m freestyle swimming; 400 mFS, 400 m freestyle swimming.

**Table 3 ijerph-19-04463-t003:** Descriptive statistics and differences derived by analysis of variance (ANOVA) among qualitative groups for perimeter players (results are presented as means ± standard deviations).

	Low Achievers	Medium Achievers	High Achievers	ANOVA		
	*N* = 23	*N* = 12	*N* = 15	F-Test	*p*	η^2^
Body height (cm)	182.81 ± 4.63	181.31 ± 5.63	187 ± 3.77	5.52	0.01	0.22
Body mass (kg)	78.71 ± 7.99	81.71 ± 5.63	81.88 ± 6.17	1.17	0.32	0.04
BMI (kg/m^2^)	23.54 ± 2.07	24.85 ± 1.05	23.4 ± 1.36	2.04	0.14	0.08
Arm span (cm)	191.03 ± 6.84	188.68 ± 7.4	193.53 ± 4.8	1.61	0.21	0.06
Body fat (%)	19.05 ± 3.98	18.18 ± 2.96	17.57 ± 2.95	0.59	0.56	0.13
25 mFS (s)	13.27 ± 0.69	12.81 ± 0.64	12.14 ± 0.31	5.01	0.03	0.21
15 mWS (s)	9.08 ± 0.38	8.85 ± 0.26	8.81 ± 0.42	2.05	0.12	0.08
100 mFS (s)	62.94 ± 3.71	60.51 ± 2.01	58.51 ± 2.44	9.68	0.001	0.29
400 mFS (s)	308.65 ± 21.26	290.32 ± 12.66	281.54 ± 14.62	11.24	0.001	0.32

Legend: BMI, body mass index; 25 mFS, 25 m freestyle swimming; 15 mWS, 15 m water polo swimming; 100 mFS, 100 m freestyle swimming; 400 mFS, 400 m freestyle swimming.

## Data Availability

Authors will provide data to all interested parties upon reasonable request.

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
