# Peer review of "Validity of the Swimming Capacities and Anthropometric Indices in Predicting the Long-Term Success of Male Water Polo Players: A Position-Specific Prospective Analysis over a Ten-Year Period"

_ijerph, 2022, doi:10.3390/ijerph19084463_

Round 1

Reviewer 1 Report

Congratulations to the authors for conducting the study, organizing the manuscript, and in-depth discussion of the results. I understand the need for some corrections and changes:

In the final of the first Introduction paragraph, are those information necessary? Do they contribute to the study? (lines 44 - 47)

Line 134: Are the "n" missing ? (xy)?

Lines 184 - 188: I suggest to inform the specific "n" in each group.

Lines 192: To calculate and present in the Tables the mean limits of confidence are mandatory, as, in the Tables, the eta2 values regarding the ANOVAs comparisons

Line 299:  What is "several" if just three results are highlighted?

Author Response

Reviewer 1

Congratulations to the authors for conducting the study, organizing the manuscript, and in-depth discussion of the results. I understand the need for some corrections and changes:

RESPONSE: Thank you for recognizing the quality of our work, and thank you for your comments. We followed it and amended the manuscript accordingly

In the final of the first Introduction paragraph, are those information necessary? Do they contribute to the study? (lines 44 - 47)

RESPONSE: Indeed, this part of information was not necessary, so we omitted it. Thank you.

Line 134: Are the "n" missing ? (xy)?

RESPONSE: Thank you for noticing it. "XY" is replaced with exact numbers, and text now reads "For the purpose of this study, participants were observed according to their playing positions and were divided in two groups, (i) center players (center forwards and points, n=35) and (ii) perimeter players (wings and drivers, n=50). " Please see 1st paragraph of the Participants subsection

Lines 184 - 188: I suggest to inform the specific "n" in each group.

RESPONSE: "N" is specified for each group and text now reads: "The first group consisted of those players who were team members of senior-level water polo teams that competed in international-level competition (G1; high achievers; n = 29). In the second group, we clustered those players who competed at the senior level in teams, which competed in the highest national playing division (G2; medium achievers; n = 26). The third group consisted of players who could not be categorized in G1 and G2 but were screened initially as G1 and G2 (low achievers, n = 30). " Please see 4th paragraph of the variables subsection

Lines 192: To calculate and present in the Tables the mean limits of confidence are mandatory, as, in the Tables, the eta2 values regarding the ANOVAs comparisons

RESPONSE: Thank you for your suggestion. If we understood it properly you have asked for adding partial eta squared values (ANOVA effect size) in the Tables? It is now added in the last column of the Tables 1-3, and presented briefly in the Results section (and previously in Statistics subsection of the Methods).

Text in Statistics now reads: "Differences among qualitative groups (high achievers – medium achievers – low achievers) were evidenced throughout one-way analysis of variance (ANOVA) with Scheffe’s post-hoc comparison. The effect size differences (ES) were evidenced on the basis of ANOVA derived partial eta squared values (η2; small ES: >0.02; medium ES: >0.13; large ES: >0.26)."

In Results section we added details about ES, such as: "The ES differences among groups were evidenced as being medium for body height, 25mFS, 15mWS, 100mFS, and 400mFS, and small for remaining variables." (for total sample); "with large ES differences among groups for body height and 100mFS, and medium ES differences for arm span and 400mFS." (for Centers); and "The ES differences among groups were evidenced as being of large magnitude for 100mFS and 400mFS, and of medium magnitude for body height and 25mFS" (for perimeter players).

Line 299:  What is "several" if just three results are highlighted?

RESPONSE: Thank you for noticing it. Text is amended and now reads: "With regard to study aims, there are three most important findings. " (please see 1st sentence of the Discussion section).

Staying at your disposal!

Authors

Reviewer 2 Report

The present study investigates the role of swimming capacities and anthropometric indices in predicting water polo success over ten years. The manuscript is well designed, presented, and written, and there are only some minor concerns needing an explanation. 

First, I would like to know why the authors chose multinomial regression instead of ordinal regression. Although the authors did not violate any assumptions, they may not be answering the research question, considering that the dependent variable is ordinal. I suggest reconsidering the statistical approach. I don't believe that changing from multinomial to ordinal will dramatically affect the results.

Secondly, performing different analyses on one sample and its subsample may inflate the P-value. So, probably, you should adjust it for multiple comparisons at the sub-sample level. 

The 15-meter water polo sprint is abbreviated in different ways. Could you please check it throughout the text?

For the last test, they couldn't push themselves from the wall, if I understood. Maybe "not able" could be misleading.

Author Response

Reviewer 2

The present study investigates the role of swimming capacities and anthropometric indices in predicting water polo success over ten years. The manuscript is well designed, presented, and written, and there are only some minor concerns needing an explanation. 

RESPONSE: Thank you for recognizing the quality of our work and for your suggestions. We have followed it strictly, so please find responses and amendments as it follows.

First, I would like to know why the authors chose multinomial regression instead of ordinal regression. Although the authors did not violate any assumptions, they may not be answering the research question, considering that the dependent variable is ordinal. I suggest reconsidering the statistical approach. I don't believe that changing from multinomial to ordinal will dramatically affect the results.

RESULTS. Indeed, there are possible controversies about our statistical approach, and by all means - we could use "standard" logistic regression in our study. However, on the basis of our previous experience (and suggestions of some other reviewers in some of our previous works), with binomial approach appears a problem of dividing subjects who are very similar in some dependent variable into separate groups. In our case, it actually means that if we decide to divide players into two groups (no matter if "medium achievers" will be observed as "high-level" or "low-level group"), this problem will still exist. In other words, it would be  possible that players divided into opposed groups will be very similar in their status. Being honest, we calculated the logistic regressions as well, but found out multinomial approach as being more suitable in evidencing the "real-sport" predictors of the achievement in studied players. Hope that you will accept our explanation, but if you will insist on calculating binomial logistic regression, we are staying at your disposal. For a moment, we briefly presented this problem in "limitation" section and text reads: "As a methodological remark we must mention that there is a possibility that some other statistical approaches were also applicable (i.e. dividing players into only two performance groups, and consequent calculation of the logistic regression), but we were of the opinion that the applied clustering into three groups and calculation of the multinomial regression allowed us to identify the “real-sport” indicators of success more specifically."

Secondly, performing different analyses on one sample and its subsample may inflate the P-value. So, probably, you should adjust it for multiple comparisons at the sub-sample level. 

RESPONSE: Thank you for your suggestion. We must agree that p-values of the ANOVA results for total sample could be "inflated" due to larger sample of subjects, but on the basis of the suggestion of 1st Reviewer, in this version of the manuscript we also included partial eta squared values as effects size difference indicator. Also, as you can see we weren't even focused on differences in total sample, but discussed profoundly only predictors of achievement for specific playing positions. The differences among performance groups in total sample (as well as predictors of achievement) were presented and discussed only sporadically since previous studied regularly used such methodological approach (personally, we don't think that studying predictors for total sample is correct approach since positions in water polo at advanced performance level are  strictly defined and therefore position-specific approach is definitively more applicable with regard to ecological validity and applicability of the results.

The 15-meter water polo sprint is abbreviated in different ways. Could you please check it throughout the text?

RESPONSE: Thank you, we checked it and corrected i, the 15mWS is used consistently

For the last test, they couldn't push themselves from the wall, if I understood. Maybe "not able" could be misleading.

RESPONSE: Thank you, it is amended as suggested, and text now reads: "For the 15mWS, the participants started from the pool, 15 meters from the finishing block, and they couldn’t push themselves from the wall." (please see Variables subsection - highlighted text)

Staying at your disposal!

Reviewer 3 Report

This is a very interesting manuscript on the evaluation of the swimming and anthropometric tests towards the prediction of long-term success in water polo. The introduction explains the problem adequately with sufficient and up-to-date references, while the methodology gives sufficient information for reproducing the findings. The discussion section clearly describes the significance of the findings in relation to what was already known. Overall, this manuscript is written exceptionally well.

Author Response

This is a very interesting manuscript on the evaluation of the swimming and anthropometric tests towards the prediction of long-term success in water polo. The introduction explains the problem adequately with sufficient and up-to-date references, while the methodology gives sufficient information for reproducing the findings. The discussion section clearly describes the significance of the findings in relation to what was already known. Overall, this manuscript is written exceptionally well.

RESPONSE: All we can say is thank you for sour support. We tried to do our best, and as we can see - you recognized our efforts. Thank you once again.